# Genetic regulation of methylation across East Asian and European populations

Ruize Liu[1,2,14], Tzu-Ting Chen[3,14], Yan Xia[4,14], Shu-Chin Lin[3,5], Tian Ge[2,6], Chia-Yen Chen [7,13] ✉, Yen-Chen Anne Feng [8,9] ✉, Hailiang Huang [1,2,10] ✉ & Yen-Feng Lin [3,11,12] ✉

Methylation quantitative trait loci (mQTL) studies have predominantly focused on European populations (EUR), limiting understanding of the genetic regulation of DNA methylation in other populations. We conduct an East Asian (EAS) mQTL analysis, integrating data from three independent samples comprising 7619 Han Chinese individuals. We identified 331,048 mCpGs, including 28,978 novel mCpGs in EAS. While observing substantial sharing of mQTL between EUR and EAS, we also identify EAS-specific mQTLs, often driven by variants with low minor allele frequencies in EUR. We found that mQTLs enriched for disease and trait heritability, especially for matched-ancestry mQTLs, underscoring their utility for interpreting GWAS results and highlighting the role of DNA methylation in diseases. Our EAS mQTL resource provides valuable insights into the genetic architecture of DNA methylation and its contribution to complex traits.

DNA methylation (DNAm), a crucial epigenetic modification, plays a vital role in regulating gene expression and has been implicated in a wide range of human diseases and traits, including cancer, cardiovascular disease, and neurodevelopmental disorders[1]. DNAm is regulated by genetic factors, with an average heritability range of 0.1–0.3[2]. Understanding the genetic factors influencing DNAm is critical for unraveling the complex interplay between genetics and epigenetics in health and disease.

Methylation quantitative trait loci (mQTLs) are genetic variants associated with DNAm levels, and their identification through mQTL mapping provides valuable insights into the genetic architecture of interindividual epigenetic variation[3]. While mQTL studies have successfully identified numerous cis-acting mQTLs, most have focused on

the European (EUR) populations[4,5]. This bias limits the generalizability of findings to other populations and hinders our understanding of disease-associated genetic loci outside of the EUR populations. Recent studies have highlighted the importance of studying diverse populations, revealing distinct population-specific regulatory mechanisms[6–8]. For instance, a previous study has identified differentially methylated sites associated with type 2 diabetes (T2D) and diabetic kidney disease in the East Asian (EAS) population[9].

Despite increasing efforts to characterize mQTLs in diverse populations[6,10–12], EAS mQTL studies remain limited in several aspects. First, even the largest EAS mQTL study to date[12] is considerably smaller (tenfold) than the largest EUR mQTL study[5], underscoring the need for larger sample sizes in EAS to achieve comparable statistical power.

[1]Analytic and Translational Genetics Unit, Department of Medicine, Massachusetts General Hospital, Boston, MA, USA. [2]Stanley Center for Psychiatric Research, The Broad Institute of MIT and Harvard, Cambridge, MA, USA. [3]Center for Neuropsychiatric Research, National Health Research Institutes, Miaoli, Taiwan. [4]Department of Molecular Biophysics and Biochemistry, Yale University, New Haven, CT, USA. [5]Institute of Statistics and Data Science, National Taiwan University, Taipei, Taiwan. [6]Psychiatric and Neurodevelopmental Genetics Unit, Center for Genomic Medicine, Massachusetts General Hospital, Boston, MA, USA. [7]Biogen, Cambridge, MA, USA. [8]Institute of Health Data Analytics and Statistics, College of Public Health, National Taiwan University, Taipei, Taiwan. [9]Department of Public Health, College of Public Health, National Taiwan University, Taipei, Taiwan. [10]Department of Medicine, Harvard Medical School, Boston, MA, USA. [11]Department of Public Health & Medical Humanities, School of Medicine, National Yang Ming Chiao Tung University, Taipei, Taiwan. [12]Institute of Behavioral Medicine, College of Medicine, National Cheng Kung University, Tainan, Taiwan. [13]Present address: Merck & Co. Inc., Cambridge, MA, USA. [14]These authors contributed equally: Ruize Liu, Tzu-Ting Chen, Yan Xia. ✉e-mail: chiayenc@gmail.com; ajfeng@ntu.edu.tw; hhuang@atgu.mgh.harvard.edu; yflin@nhri.edu.tw

Second, some EAS mQTL studies have used older platforms, such as the Illumina HumanMethylation450K array (450K), which offers less CpG coverage compared to the newer Infinium HumanMethylatio-nEPIC array (EPIC), potentially constraining novel mQTL discovery[11]. Third, the extent to which mQTLs identified in one population can be used to annotate and interpret trait and disease genetic components in other populations (cross-population transferability) remains unclear, and such investigation is limited by the sample size and technology gaps mentioned above.

To address these limitations, we conducted a large-scale mQTL study ($n = 7619$), including newly analyzed data from the Taiwan Bio-bank ($n = 1997$) and two recently published mQTL studies of Han Chinese samples ($n = 2099$ and $3523$)[11,12]. These combined samples effectively double the sample size of previous EAS mQTL studies. By leveraging the increased sample size and the expanded CpG coverage of the EPIC array, our study aims to identify novel EAS mQTLs and provide crucial insights into the cross-population transferability of mQTL contributions to the genetic components of complex traits and diseases. This work bridges the gap in our understanding of DNAm regulation across populations and clarifies how genetic and epigenetic variation influence health and disease.

## Results

### Study sample

We analyzed data from both EAS and EUR populations. The EAS mQTL analysis included 7619 subjects from three samples: the Taiwan Bio-bank (EAS_TWB) and two published studies (EAS_Peng and EAS_Hatton)[11,12]. EAS_TWB and EAS_Peng quantified DNAm levels at approximately 800 K CpG sites using the EPIC array, whereas EAS_Hatton assessed DNAm at approximately 400 K CpG sites using the 450K array. All three EAS studies had genotype data for 7.3−8 million genetic variants. The EUR mQTL analysis comprised 27,750 individuals from the EUR_Min study[5], which measured DNAm levels at approximately 420 K CpG sites using the 450K array and provided genotype data for approximately 10 million genetic variants (Fig. 1).

### mQTL and mCpG sites in Taiwan Biobank

Approximately 8 million variants, 755,476 CpG sites, and 1,997 subjects were available from TWB after stringent quality control (QC) analyses (see "Methods" section). Using QTLtools[13], we performed association tests to identify cis-mQTLs, defined as genetic variants within ±1 Mb of each CpG site, that are associated with their DNAm levels (see "Methods" section). We identified 36,048,502 cis-mQTLs with $P$-value < 1e-11 (Bonferroni corrected significance threshold by total number of tests = 0.05/4,149,372,847 = 1.2e-11), implicating 191,428 mCpG sites and 4,890,720 variants. Following a previously established definition[12], we defined mCpG as a CpG site whose methylation levels were significantly associated with at least one genetic variant.

The median distance between the most associated variant and its corresponding mCpG site was 7.3 kb (Fig. 2a). Using the heritability estimates from a twin study conducted in the Netherlands Twin Reg-ister (NTR) biobank[14,15], CpG sites with mQTLs (mCpGs) we identified have significantly higher median heritability than those without (median $h^2 = 39\%$ vs 8%, two-sided Mann–Whitney $U$-test, $P$-value < 2.2e-16; Fig. 2b). We observed that as the heritability of CpG sites increased, the ratio of mCpGs to non-mCpGs also increased (Fig. 2b), even though the heritability estimates of CpG sites were derived from different populations. This finding is consistent with our expectation that mCpGs are under larger genetic influences than non-mCpGs and further suggests that the mQTL regulatory mechanism may be shared across populations.

We further investigated methylation features of mCpGs in the EAS_TWB. A greater proportion (62.9%) of mCpGs exhibited inter-mediate methylation levels (20% < mean DNAm < 80%), whereas non-mCpGs tended to cluster at either low (<20%) or high (>80%) DNAm levels (Fig. 2c). We then performed enrichment analyses on mCpGs using CpG categories and chromatin states from 127 cell types to gain further insights into the regulatory landscape of mCpGs. Our analysis revealed that mCpGs were significantly enriched in N_Shore (OR = 1.03, $P$-value = 7e-4) and S_Shore (OR = 1.06, $P$-value = 6e-10) relative to non-mCpGs, while mCpGs were depleted in Island (OR = 0.4,

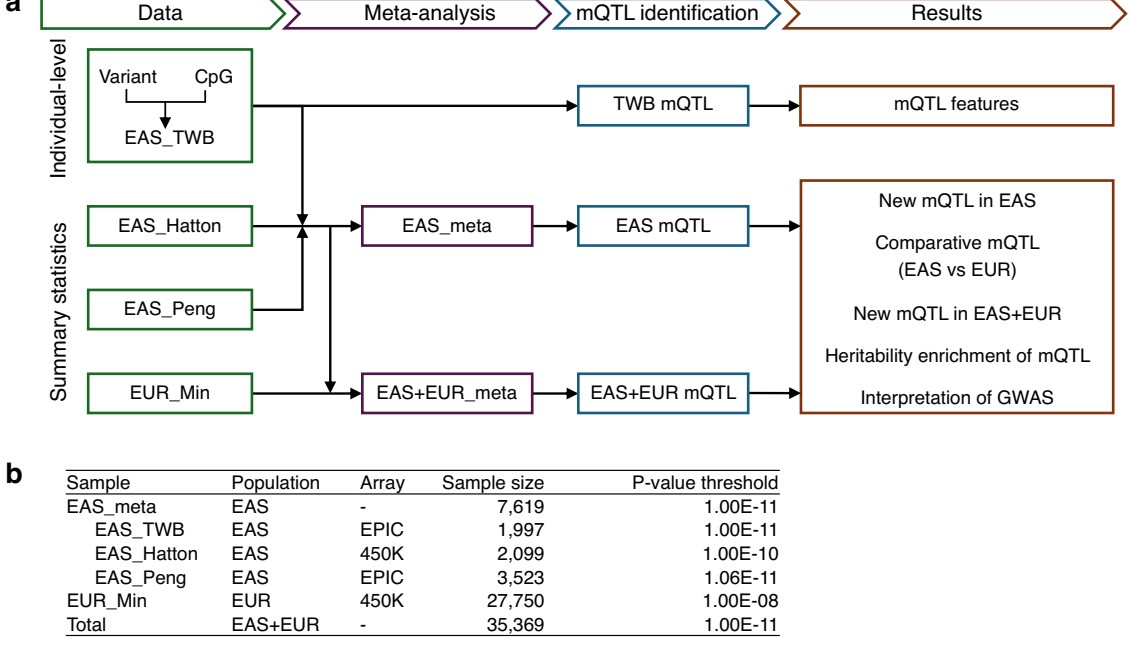

**Fig. 1 | Overview of the study design. a** Data and analyses in this study. **b** Sample information. *P*-value thresholds for previous mQTL studies (EAS_Hatton, EAS_Peng, and EUR_Min) were adopted as reported in the original publications[5,11,12]. The *P*-value threshold for this study was described in the "Method" section. The *P*-values were derived from two-sided statistical tests, without correction for multiple comparisons.

| Sample | Population | Array | Sample size | P-value threshold |
|---|---|---|---|---|
| EAS_meta | EAS | - | 7,619 | 1.00E-11 |
| EAS_TWB | EAS | EPIC | 1,997 | 1.00E-11 |
| EAS_Hatton | EAS | 450K | 2,099 | 1.00E-10 |
| EAS_Peng | EAS | EPIC | 3,523 | 1.06E-11 |
| EUR_Min | EUR | 450K | 27,750 | 1.00E-08 |
| Total | EAS+EUR | - | 35,369 | 1.00E-11 |

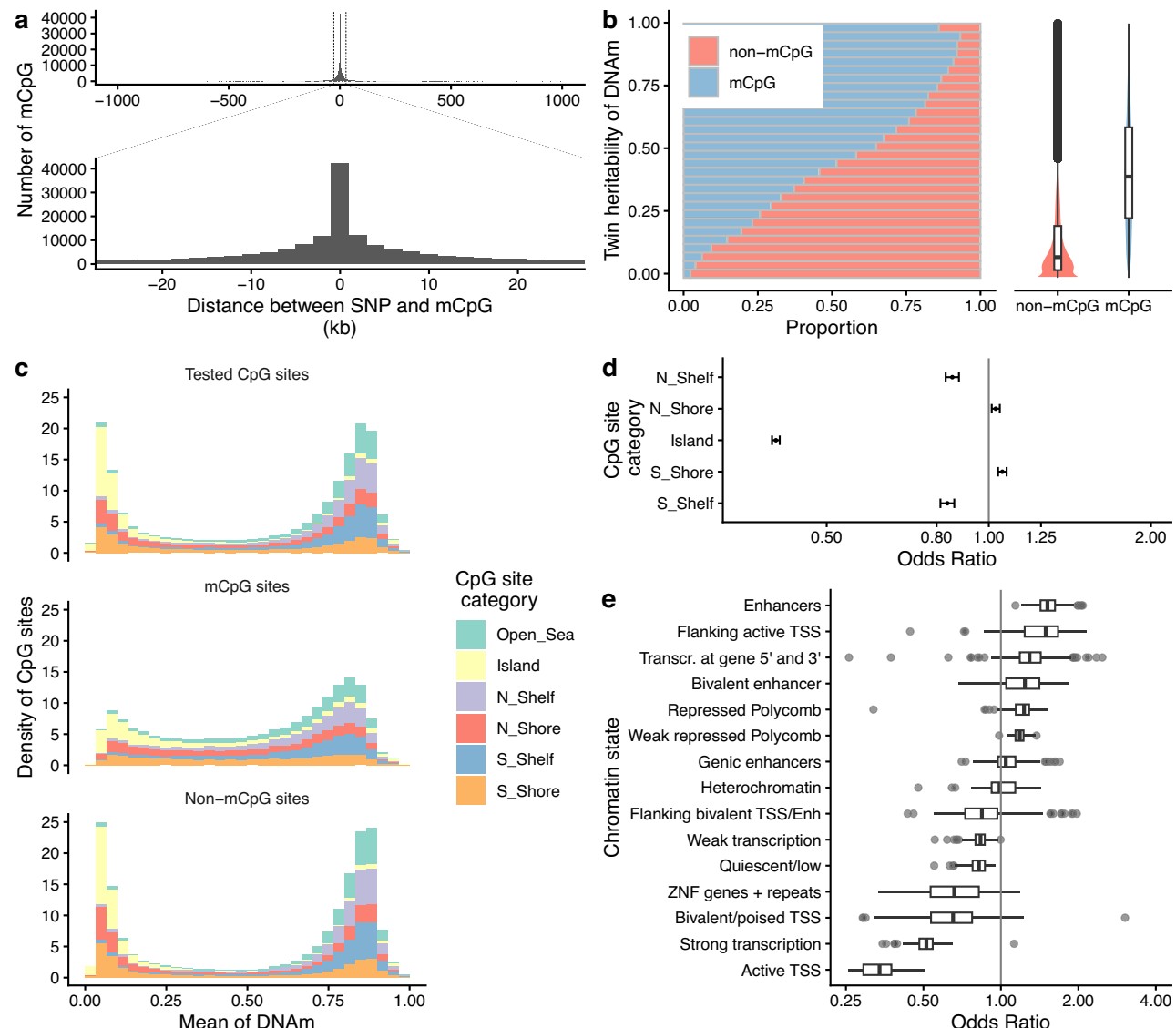

**Fig. 2 | Features of mCpG in Taiwan Biobank. a** Distribution of the distance between the most associated variants and mCpG. The distance is calculated as (the genome position of the variant - the genome position of its corresponding CpG site). kb kilobase. **b** Relationship between CpG site heritability and the proportion of mCpG. Comparison of heritability between mCpG and non-mCpG. Heritability estimates for 364,047 CpG sites are extracted from a previously published twin study[14]. Both 83,060 and 280,987 CpGs are in mCpG and non-mCpG groups, respectively. **c** Distribution of DNAm levels across CpG site categories. **d** Enrichment of mCpG in CpG site category relative to non-mCpG, using the open sea as the background. The odds ratio was estimated for Island (CpG island,

$n$ = 579,478 CpG sites), N_Shelf (CpG north shelf, $n$ = 459,389 CpG sites), N_Shore (CpG north shore, $n$ = 507,576 CpG sites), S_Shelf (CpG south shelf, $n$ = 457,379 CpG sites), S_Shore (CpG south shore, $n$ = 496,514 CpG sites). The dot represents the odds ratio, and the error bar represents the 95% confidence interval. The $X$-axis shows the odds ratio on a log scale. **e** Enrichment of mCpG in chromatin states relative to non-mCpG ($n$ = 127 cell types). Chromatin states: chromHMM states across 127 cell types. The $X$-axis shows the odds ratio on a log scale. In the boxplot of (**b**, **e**), center line: median; box limits: upper and lower 25% quartiles; whiskers: 1.5× interquartile range; point: outliers. Source data are provided as a Source Data file.

$P$-value < 1e-50), N_Shelf (OR = 0.85, $P$-value = 7e-28), and S_Shelf (OR = 0.83, $P$-value = 7e-33; Fig. 2d; Supplementary Data 1). Across chromatin states, mCpGs were enriched in enhancers (median OR = 1.52, $P$-value = 1.8e-4) and weak repressed polycomb regions (median OR = 1.18, $P$-value = 1.2e-3; Fig. 2e and Supplementary Data 2), relative to non-mCpG. Conversely, mCpG were depleted in active transcription start sites (TSS) (median OR = 0.30, $P$-value = 4.2e-6) and strong transcription regions (median OR = 0.51, $P$-value = 6.2e-8; Fig. 2e and Supplementary Data 2).

Compared with previous studies, we found the features of mQTLs in the EAS_TWB were highly consistent with previously identified cis-mQTLs. Specifically, regulatory cis-mQTL variants tend to be located

proximally to their associated CpG sites, and mCpGs are enriched in enhancer regions, N_Shore and S_Shore, relative to non-mCpGs[5,12]. Furthermore, comparison of our EAS_TWB mCpG results with those from two previous EAS studies (EAS_Hatton and EAS_Peng) revealed a high degree of replication. Specifically, 182,622 (95%) of our identified mCpGs were also reported as significant mCpGs in at least one of the two prior studies, based on their respective significance thresholds (Fig. 1b and Supplementary Fig. 1). A substantial overlap was observed with EAS_Peng, which used the EPIC array, where 93.89% of EAS_TWB mCpGs were replicated. Among these, 53.18% were exclusively detected by the EPIC array, highlighting the increased coverage of this platform. In contrast, 37.18% of EAS_TWB mCpGs were replicated in

EAS_Hatton, which used the 450K array and captured substantially fewer CpG sites. This cross-study replication of mCpGs and their features suggests the high quality of the TWB data and the reliability of mQTL analysis, while providing new mCpGs and mQTLs from an independent data source.

## mQTL discovery in EAS and across EAS and EUR

As the mQTL effect sizes are highly consistent across the three EAS studies (EAS_TWB, EAS_Hatton, and EAS_Peng; $r > 0.9$, Supplementary Fig. 2, Methods), we performed a meta-analysis using a fixed-effect model to improve the power for mQTLs discovery in EAS. This analysis identified 88,092,782 significant mQTLs ($P$-value < 1e-11) regulating the methylation level of 331,048 CpG sites (mCpGs). The EAS meta-analysis (EAS_meta) identified an additional 43,313 mCpGs (13% of total mCpGs) that were not detected in the three individual EAS mQTL studies (Supplementary Fig. 3). When comparing our EAS meta-analysis results with previously published mQTL studies in EAS and EUR (EAS_Hatton, EAS_Peng, and EUR_Min), we found 28,978 CpGs were identified as mCpGs for the first time.

We then compared EAS_meta with the largest published EUR mQTL study (EUR_Min) for insights into the extent to which mQTLs are replicable across EUR and EAS populations. We found that 127,650 (35% of EAS_meta) mCpGs were also reported in EUR_Min (Supplementary Fig. 4). Of the remaining 203,398 (65% of EAS_meta) mCpGs that were found only in EAS_meta, 159,928 (78.6%) were unique to the EPIC array, suggesting that the higher coverage of the EPIC array, which was only used by some of the EAS studies, likely contributed to the identification of mCpGs exclusive to EAS_meta. Overall, only 50% of the mCpGs identified in EAS_meta are present in both EPIC and 450K array platforms (164,702 mCpGs). Among them, 74.3% were also reported as mCpGs in EUR (EUR_Min). On the other hand, 71.5% of the mCpGs identified in EUR_Min that are present in both array platforms were also found in EAS_meta (Supplementary Fig. 4). This high degree of mCpG overlap at shared CpGs between EAS_meta and EUR_Min suggests similar genetic regulatory mechanisms of methylation in these populations.

For CpGs present on both array platforms but showed significant associations only in EAS_meta, their index variants exhibited significantly higher minor allele frequency (MAF) in EAS compared to EUR (average MAF in EAS vs EUR: 0.273 vs 0.244, two-sided paired $t$-test $P$-value = 2.14e-10, Supplementary Fig. 5). In contrast, only slight MAF differences were observed for the index variants of mCpGs shared between EAS and EUR (average MAF in EAS vs EUR: 0.272 vs 0.270, two-sided paired $t$-test $P$-value = 7.42e-6, Supplementary Fig. 5). These findings underscore the importance of including diverse populations in mQTL studies to identify population-specific genetic factors influencing DNAm.

To maximize the power for mQTL discovery, we performed a fixed-effect meta-analysis combining data from the EAS (EAS_TWB, EAS_Hatton, EAS_Peng) and EUR (EUR_Min) studies, given the high correlation of mQTL effect sizes between EAS and EUR ($r = 0.82$; Supplementary Fig. 6; Methods). This meta-analysis identified 131,491,413 mQTLs ($P$-value < 1e-11) regulating the methylation levels of 382,417 CpG sites (mCpGs). Of these mCpGs, 29,078 represented novel mCpGs not identified in the previous studies (EAS_Hatton, EAS_Peng, or EUR_Min; Supplementary Fig. 7).

We further characterized the regulatory functions of 29,084 newly identified mCpG sites identified in either EAS or cross-population (EAS and EUR) meta-analyses. Our findings indicate that these new mCpGs generally share regulatory patterns with known mCpGs, while also affecting new regulatory regions. Specifically, the proportion of new mCpGs in DNase Hypersensitivity Sites (DHSs) (57%) was similar to that of known mCpGs (60%). Likewise, we observed strong correlations in the proportion of new vs known mCpG overlaps with cell-type-specific transcription factor binding sites (TFBSs)

(Pearson $r = 0.94$) across 171 TFs and 91 cell types from ENCODE (Supplementary Fig. 8a). This consistency extended to chromatin states, where the proportion of new vs known mCpG overlaps with 15 chromatin states across 127 cell types from the Roadmap Epigenomics project also showed a high correlation (Pearson $r = 0.98$, Supplementary Fig. 8b). Among these chromatin states, the Quiescent state contained the largest proportion of both novel and known mCpG (Supplementary Fig. 8b).

Beyond these consistent patterns, our analysis revealed new regulatory regions influenced by mCpGs. For example, we identified 28 imprinting control regions (ICRs; 23 maternal and 5 paternal) overlapping with new mCpGs, with 12 of these (8 maternal and 4 paternal) being newly identified mCpG-ICR overlaps (Supplementary Fig. 8c). Additionally, new mCpGs were found at 977 CTCF binding sites, of which 835 were identified for the first time as overlapping with mCpGs, thus increasing the number of mCpG-regulated CTCF sites by 6.7%. Our analysis also indicated that new mCpGs may influence 182,386 additional cell-type-specific TFBSs, a 6% increase in the total number of TFBSs potentially regulated by mCpGs (Supplementary Fig. 8d). These findings suggest that the identification of these new mCpGs enhances our understanding of complex gene regulation mechanisms.

## Heritability enrichment for mQTL in EAS and across EAS and EUR

To assess the extent to which the genetic underpinnings of human complex traits are driven by mQTLs, we investigated the enrichment of heritability based on mQTL annotations using linkage disequilibrium score regression (LDSC) on 48 complex traits and diseases from Bio-Bank Japan (BBJ) with SNP-heritability > 0.03 (Methods, Supplementary Data 3). We observed that EAS_meta mQTLs had greater heritability enrichment compared to most functional regions, including TSS and coding regions (Fig. 3a), suggesting mQTLs represent an important mechanism underlying the genetic contribution to complex traits and diseases.

As functional genomic resources, including mQTLs, were predominantly generated from individuals of European ancestry, we next investigated whether heritability enrichment is affected by a mismatch between the ancestries of mQTL and GWAS. Using LDSC, we compared the heritability enrichment of mQTL identified in EAS (EAS_meta) against those identified in EUR (EUR_Min) using complex traits and diseases from BBJ. As expected, EAS mQTLs demonstrated significantly higher heritability enrichment than EUR mQTLs (average enrichment: 8.97 in EAS and 6.03 in EUR, two-sided $t$-test $P$-value = 1e-6; Fig. 3b). To mitigate potential biases due to methylation array platform differences, we restricted the analysis to shared CpGs between EPIC and 450K arrays and obtained similar results (average enrichment: 7.82 in EAS and 6.02 in EUR, two-sided $t$-test $P$-value = 0.006; Fig. 3b). We further included two complex disease GWAS from consortia that are well-powered in EAS in the LDSC analysis: schizophrenia (SCZ) and inflammatory bowel diseases (IBD), including Crohn's disease (CD) and ulcerative colitis (UC)[16,17]. Consistent with the biobank findings, we found significant heritability enrichment using mQTLs from EAS (Fig. 3c, total number of tests = 8, $P < 0.05/8 = 6.3e-3$, Bonferroni correction) but not in EUR mQTLs. These findings highlight how mQTL resources from diverse ancestries may facilitate interpreting the GWAS findings.

Further, we investigated the factors driving the observed differences in mQTL enrichment in disease across EAS and EUR. Building upon previous research highlighting the influence of allele frequency differences between populations on the genetic architecture differences of complex diseases between populations[16,17], we assessed the impact of MAF differences on heritability enrichment of mQTL by calculating the difference in MAF (ΔMAF) between EAS and EUR for each variant and incorporating ΔMAF as an additional annotation in the LDSC model. We quantified the conditional annotation's

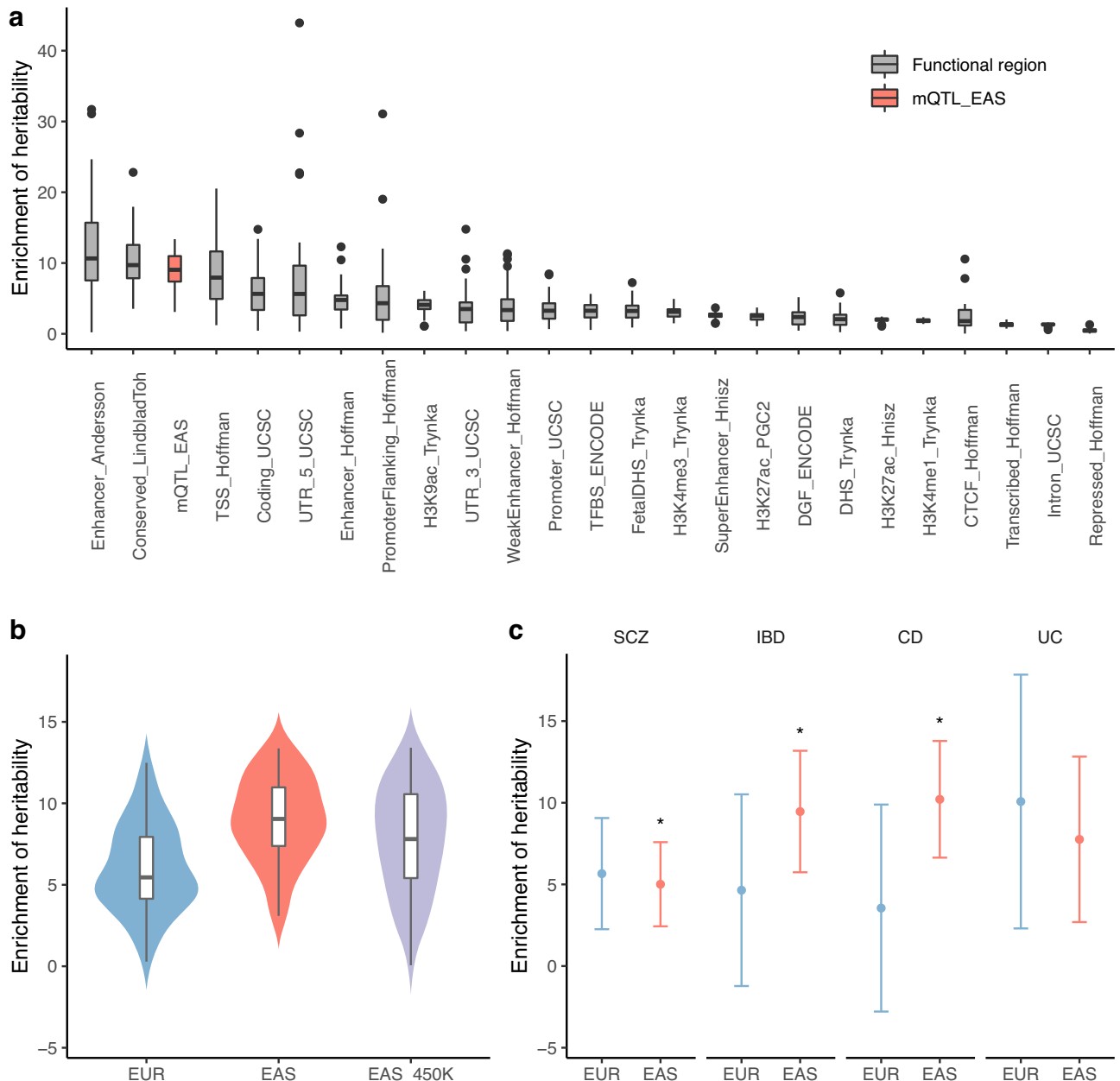

**Fig. 3 | Heritability enrichment of mQTL for complex traits. a** Enrichment of heritability for 48 complex traits and diseases from BBJ in EAS mQTL (mQTL_EAS) and genomic functional annotations (*n* = 48 traits and diseases). **b** Enrichment of heritability for 48 complex traits and diseases from the BBJ in EAS and EUR mQTL (*n* = 48 traits and diseases). EAS: EAS mQTL; EUR: EUR mQTL; EAS_450K: EAS mQTL with CpGs shared in both EPIC and 450K arrays. **c** Enrichment of heritability for IBD and SCZ GWAS from EAS in mQTL of EAS and EUR. *P*-values for heritability enrichment of EAS mQTL are 2.6e-3 (SCZ), 1.69e-5 (IBD), 1.75e-6 (CD), and 7e-3 (UC).

The corresponding *P*-values for EUR mQTLs in IBD and SCZ are 7e-3 (SCZ), 0.21 (IBD), 0.4 (CD), and 0.02 (UC). The *P*-values are derived from two-sided statistical tests within the partitioned LD score regression, without correction for multiple comparisons. *: significant enrichment (total number of tests = 8, *P* < 0.05/8 = 6.3e-3, Bonferroni correction). The dot represents enrichment of heritability, and the error bar represents the 95% confidence interval. In the boxplot of (**a**, **b**), center line: median; box limits: upper and lower 25% quartiles; whiskers: 1.5× interquartile range; point: outliers. Source data are provided as a Source Data file.

contribution to disease heritability using the standardized effect size ($\tau_*$, see "Method" section) for traits in BBJ, SCZ, and IBD (including CD and UC). In BBJ, the difference in standardized effect sizes ($\Delta\tau_*$) between EAS and EUR significantly decreased after conditioning on $\Delta$MAF (average $\Delta\tau_*$ from the LDSC model without $\Delta$MAF vs with $\Delta$MAF: 2.90 vs 2.25; *P*-value = 3.83e-22, two-sided paired *t*-test), indicating that MAF differences partially contribute to the population-specific effects of mQTLs, which in turn affect their disease heritability enrichment. When focusing on CD, UC, IBD, and SCZ, we observed a similar pattern

that $\Delta\tau_*$ decreased from 3.9 to 3.6 for CD, 1.19 to 0.4 for UC, 3.4 to 3.2 for IBD, and 0.97 to 0.2 for SCZ after conditioning on $\Delta$MAF (*P*-value = 0.03, two-sided paired *t*-test). The results demonstrate that MAF differences across populations influence the cross-population transferability of mQTL contributions to complex disease heritability.

### mQTL for causal interpretation of complex trait GWAS in EAS
To identify mCpG sites with potential causal effects on complex traits and diseases, we performed Hypothesis Prioritization for multi-trait

Colocalization (HyPrColoc), followed by validation using summary data-based Mendelian Randomization (SMR). We used EAS mQTLs (EAS_meta) and GWAS summary statistics for 220 traits and diseases from BBJ, along with summary statistics for SCZ and IBD (including CD and UC) from consortia in EAS.

We found 9792 CpG-trait clusters in the HyPrColoc analysis (posterior probability of colocalization [PP] > 0.7) involving 11,465 CpG-trait pairs (Supplementary Data 4). Among them, 339 CpG-trait pairs were also identified in the SMR analysis (P_SMR < 5e-10, Bonferroni correction, and P_HEIDI > 0.05; Supplementary Data 5 and 6). For our downstream analysis, we conservatively focus our findings on the 339 significant CpG-trait pairs that are shared between both analyses (Supplementary Data 6 and Fig. 4).

We identified cg10771262, located in the promoter region of *TCF21*, exhibiting pleiotropic effects on coronary artery diseases (CAD; PP = 0.95, Fig. 5a), including angina pectoris (Angina), unstable angina pectoris (UAP), stable angina pectoris (SAP), and myocardial infarction (MI), as well as the usage of related medications, including vasodilators used in cardiac diseases (ATC_C01D) and salicylic acid derivatives (ATC_N02BA) and white blood cell count (WBC) in EAS. HyPrColoc identified rs2327429 as a candidate causal variant explaining 100% of PP for the shared association signal across cg10771262 DNAm and the 7 traits. SMR analysis further confirmed the significant associations between cg10771262 and SAP (P_SMR = 1.39e-13), MI (P_SMR = 2.69e-12), and ATC_C01D (P_SMR = 3.2e-11), respectively. Based on the SMR result, we observed that increased cg10771262 DNAm levels were positively associated with the risk of CAD (angina, UAP, SAP, and MI) and the use of drugs (ATC_C01D and ATC_N02BA), while negatively associated with WBC (Supplementary Data 5). In a previous study, murine models have shown that *Tcf21* is essential for the phenotypic modulation of smooth muscle cells in atherosclerotic tissues, promoting a fibroblast phenotype[18]. In humans, *TCF21* expression has been linked to a reduced risk of CAD[19], and rs2327429, located in the promoter region of *TCF21*, has been associated with CAD risk with a posterior probability of 1 in the fine-mapping analysis[20]. The C allele of rs2327429 is associated with increased *TCF21* gene expression in the left ventricle of the heart (P-value = 4.3e-6) and the atrial appendage of the heart (P-value = 2.78e-5) in GTEx v8 eQTL analysis. In our analysis, the C allele of rs2327429 was associated with decreased methylation at cg10771262 and decreased risk of heart-related diseases. It suggests that the C allele of rs2327429 may increase *TCF21* expression by reducing the methylation level of cg10771262 in the *TCF21* promoter region, thereby decreasing the risk of heart-related diseases (Fig. 5b).

Additionally, we replicated a previously reported association between CpG sites in *CAMK1D* and T2D[21]. Specifically, our results demonstrate colocalization of cg10704395, cg03575602, and cg14427642, in the promoter region of *CAMK1D*, with a group of diabetes-related traits including drugs used in diabetes (ATC_A10), body mass index (BMI), birth weight (BW), cataract, glucose levels, hemoglobin A1c (HbA1c), type 1 diabetes (T1D), and T2D (PP > 0.9; Supplementary Fig. 9). These analyses indicate three candidate causal variants in high LD (rs11257655, rs11257657, and rs4747971, pairwise $R^2$ > 0.9). SMR analysis further confirmed the significant association of cg10704395, cg03575602, and cg14427642 with HbA1c (P_SMR = 9.57e-11, 6.63e-11, 3.41e-10, respectively) and cg03575602 with ATC_A10 (P_SMR = 1.52e-25). Based on the direction of the effect of CpGs on T2D-related traits or diseases estimated in the SMR analyses, we observed that increased CpG DNAm levels were negatively associated with levels of glucose and HbA1c, the risk of T2D, T1D, and cataract, and the use of antidiabetic drugs (ATC_A10), while positively associated with BMI and BW (Supplementary Data 5). Our results align with previous studies suggesting that rs11257655, located in an enhancer region, influences methylation of the *CAMK1D* promoter, thereby affecting *CAMK1D* expression and potentially modulating T2D risk[21].

Furthermore, EAS mQTLs provide a unique opportunity to interpret EAS-specific genetic findings, which may have been missed in EUR studies. For example, we identified four CpG-trait pairs where the CpGs were significant only in EAS, despite those CpGs being tested in both EAS and EUR (Supplementary Fig. 10a–d). The candidate causal variants of these pairs showed higher MAF in EAS compared to EUR (average MAF in EAS vs EUR: 26.7% vs 3.1%). This difference in MAF likely explains why these mQTLs were not detected in EUR populations and underscores the importance of EAS-specific mQTLs for disease mechanism interpretation.

To further explore the biological mechanisms of pleiotropic signals identified by HyPrColoc, we performed gene set enrichment analysis using MAGMA[22] for each trait within the CpG-trait clusters (see "Method" section). We looked for GO biological process (BP) gene sets from MSigDB (v2025.1) that are significantly enriched across multiple traits in the cluster. We only tested relevant gene sets, which include the gene nearest to the mCpG (under the assumption that mCpG very often regulates the nearest gene).

The 17.5% of the CpG-traits pleiotropic clusters had at least one GO BP gene set significantly enriched in more than one trait (Supplementary Data 7). For example, for the pleiotropic effects of cg10771262 (nearest gene *TCF21*) on CAD, we found the vasculature development GO BP gene set (Renal system vasculature development, Bonferroni-corrected P-value < 0.05) significantly enriched for four of six traits in the pleiotropic group: Angina, MI, SAP, and ATC_C01D (Supplementary Fig. 11). This suggests that cg10771262, through regulating *TCF21*, may mediate its pleiotropic effects on these CAD-related traits through its role in vascular development.

Additionally, we examined cg05333014, which had pleiotropic effects on high-density lipoprotein cholesterol (HDLC) and triglycerides (TG). The nearest gene to cg05333014, *APOB*, encodes apolipoprotein B (apoB), a crucial component of lipoproteins responsible for transporting lipids (fats and cholesterol) in the bloodstream. In the enrichment analysis, both HDLC and TG showed significant enrichment (Bonferroni-corrected P-value < 0.05) in lipid metabolism pathways, revealing the shared regulatory pathways between HDLC and TG (Supplementary Fig. 12).

## Discussion

Although several studies have reported mQTLs in EAS populations, previous efforts have been limited by sample sizes that are often ten-fold smaller than those of EUR mQTL studies. In this study, we conducted the largest mQTL analysis in EAS to date, including 7619 Han Chinese individuals from three datasets, more than doubling the sample size of previous EAS mQTL studies. This enabled us to identify 331,048 mCpGs, of which 28,978 represent novel discoveries. We found a high replication rate for mCpGs previously reported in EUR, suggesting that mQTL regulatory mechanisms are largely conserved across populations. We also identified numerous EAS-specific mQTLs with significantly lower MAFs in EUR, potentially hindering their detection in EUR despite the larger sample size. These findings underscore the importance of multi-ancestry mQTL studies.

We demonstrated the utility of mQTLs in interpreting GWAS results by observing a greater enrichment of disease and trait heritability within mQTLs compared to other genomic annotations, underscoring the critical role of DNAm in disease. Furthermore, greater heritability enrichment was observed when using matched-ancestry mQTLs, indicating the influence of population-specific genetic and epigenetic factors on disease risk. Matched-ancestry mQTL analyses can therefore provide more refined insights into disease mechanisms.

Our analysis also reveals the impact of MAF differences on cross-population comparisons of mQTLs. These MAF differences affect both mQTL detection power and their contribution to disease heritability across populations. For instance, an mQTL with an effect size of

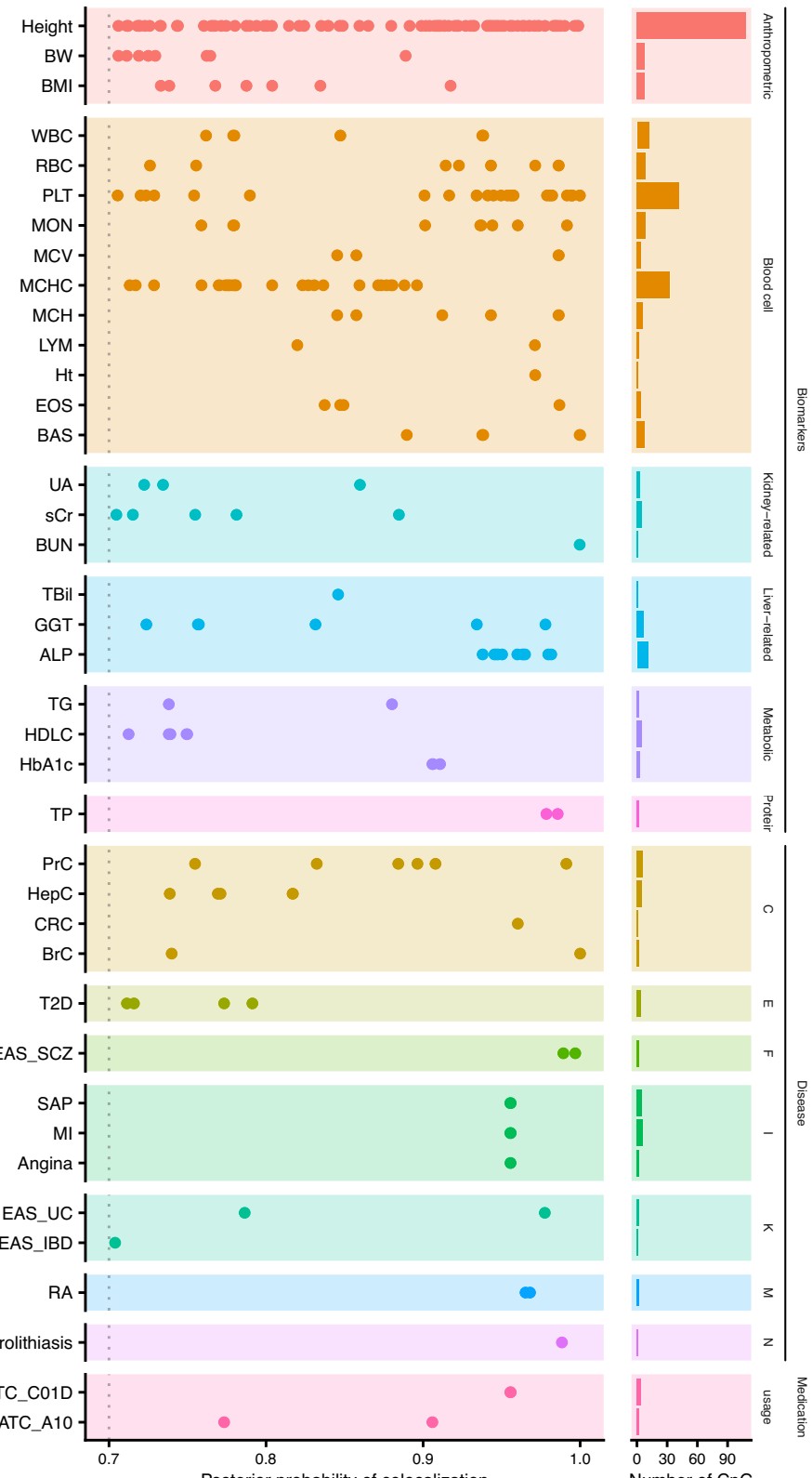

**Fig. 4 | Identification of putative causal relationships between CpG sites and complex traits/diseases.** Putative causal CpG-trait pairs were identified by overlapping the results of SMR and HyPrColoc analyses. Each dot represents each putative causal CpG-trait pair. Each row of the plot represents a single trait or disease, with traits or diseases within the same category grouped by the same color.

EAS_SCZ, EAS_UC, and EAS_IBD represent the GWAS of SCZ, UC, and IBD from consortia in EAS, respectively. The other traits or diseases were from BBJ (Supplementary Data 6). The dot lines represent the significant thresholds for HyPr-Coloc (posterior probability = 0.7). Trait abbreviations are available in Supplementary Data 6.

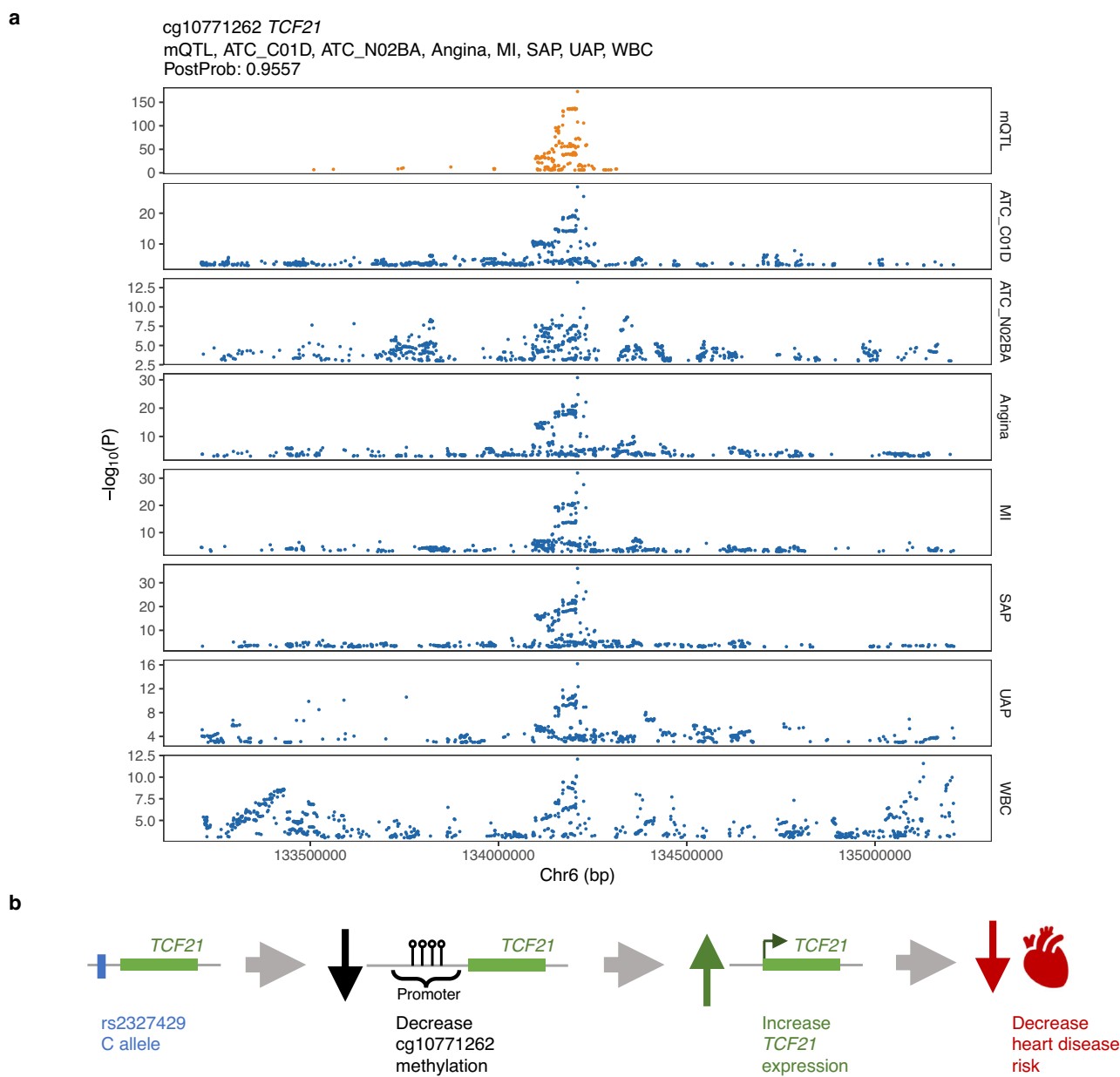

**Fig. 5 | Interpreting the molecular mechanism of a heart disease-associated genetic variant via the mQTL. a** Regional association plot of mCpG cg10771262 and its colocalized traits. mCpG cg10771262 colocalized with heart-related traits, including vasodilators used in cardiac diseases (ATC_C01D), salicylic acid derivatives (ATC_N02BA), angina pectoris (Angina), MI, SAP, UAP, and WBC, with the posterior probability of colocalization = 0.95. The nearest gene of cg10771262 is *TCF21*. Yellow: mQTL association; Blue: traits or diseases association. P: *P*-values

originate from two-sided statistical tests within the EAS mQTL meta-analysis conducted in this study or from two-sided statistical tests within the GWAS performed in BBJ, without correction for multiple comparisons. **b** Proposed molecular mechanism between genetic variation rs2327429, cg10771262 methylation, *TCF21* gene expression, and heart disease. The C allele of genetic variant rs2327429 reduces methylation at the *TCF21* promoter, leading to increased *TCF21* expression, which subsequently decreases the risk of heart disease.

1 standard deviation unit on methylation and an MAF of 0.05 in EAS and 0.001 in EUR, would require only 690 samples in EAS for 95% detection probability at a *P*-value threshold of 1e-11. In contrast, achieving the same power in EUR would require 35,680 samples, 52.3 times more than in EAS (Supplementary Fig. 13). Furthermore, differences in MAF of causal variants across populations also contribute to the heterogeneity of disease-associated loci[16,17], highlighting the influence of population-specific variants on mQTL contributions to disease.

Following best practices for interpreting GWAS results using cis-QTLs[23], we employed both SMR and HyPrColoc to investigate CpG-trait associations. While SMR assesses individual CpG-trait pairs, HyPrColoc extends this by identifying clusters of traits potentially influenced by

shared causal variants at a given CpG. This multi-trait approach provides a broader perspective on the pleiotropic effects of CpG methylation. Although not all traits within a CpG-trait cluster identified by HyPrColoc reached significance in the CpG-trait pair SMR analysis, the direction of effect estimates from SMR generally supports the inferred relationships within the cluster. For example, increased DNAm at cg10704395 was negatively associated with the risk of T2D and related traits, consistent across the identified cluster. This concordance in the direction of effect, even in the absence of statistical significance in all CpG-trait pair SMR tests, may suggest a potential shared underlying causal mechanism. However, this interpretation should be approached with caution.

Our findings also suggest that the pleiotropic effects of CpG methylation might be partially interpreted as an mCpG's influence on a gene that is involved in biological processes critical to multiple related traits. Our gene set enrichment analysis corroborated this, demonstrating that biological processes encompassing the mCpG-regulated gene were enriched in traits exhibiting pleiotropic effects. For instance, the pleiotropic effects observed for cg10771262 (nearest gene *TCF21*) on CAD-related traits (Angina, UAP, SAP, MI) and the usage of related medications (ATC_C01D and ATC_N02BA), were linked through the vasculature development biological process. This association is biologically reasonable, as these CAD-related traits represent a spectrum of ischemic heart disease, a condition primarily driven by impaired blood flow due to atherosclerosis[24,25]. Furthermore, these traits share common risk factors (e.g., high blood pressure, high cholesterol, diabetes, smoking, etc), and the related medications (e.g., vasodilators and salicylic acid derivatives) also address vascular function (e.g., widening blood vessels) and clot prevention, respectively. This strong convergence of clinical, genetic, and pharmacological evidence further supports that the pleiotropic effects on these CAD-related traits are mediated via mechanisms related to vascular development.

While our study significantly expands our understanding of mQTLs in EAS, we acknowledge its limitations. Our analysis focused on cis-mQTLs because trans-mQTL was not available in one of the constituent studies (EAS_Hatton). Consequently, our mQTL discovery, cross-population comparisons, and GWAS interpretation were restricted to cis-mQTLs. Although previous studies suggest that cis-mQTLs account for the majority (~90%) of all mQTLs[12], and thus our results likely capture a substantial proportion of the genetic influence on DNAm, future studies incorporating trans-mQTL data will provide a more comprehensive understanding of the genetic architecture of DNAm.

In summary, by leveraging the increased coverage of the EPIC array and large sample size, our study identified numerous previously unreported mQTLs in EAS, demonstrating that while regulatory mechanisms are largely shared across populations, the interpretation of genetic risk factors through mQTLs can be population-specific. In this study, we provide mQTL resources for EAS ancestry with the largest sample size to date. It enhances our understanding of DNAm regulation across diverse populations and provides valuable insights into the mechanisms underlying health and disease in EAS individuals.

## Methods

### Inclusion and ethics

This study has been approved by the Ethics and Governance Council (EGC) of Taiwan Biobank (TWBR10907-05) and the Institutional Review Board (IRB) of National Health Research Institutes, Taiwan (EC1090402-E). All relevant ethical regulations were followed.

### Sample description in TWB

**Participants.** We used data from the TWB, a government-supported prospective cohort study conducted on the Taiwanese population (https://www.biobank.org.tw/english.php). The TWB is a community-based cohort, focusing on individuals aged 20–70 with no prior cancer diagnosis. Recruitment sites were strategically placed across Taiwan, taking into account the population density of various counties and cities. During the enrollment process, participants provided written informed consent and then underwent various assessments, including questionnaires, physical examinations, and blood and urine tests. Baseline data collection was comprehensive, capturing a wide range of phenotypic measurements. It is important to mention that the TWB cohort represents a Han Chinese population and does not include data from indigenous tribes due to strict ethical and legal guidelines[26].

**Genotype data.** We obtained genome-wide genotype data from 27,719 samples in the TWBv1 custom array. The TWBv1 array, designed on the basis of the Thermo Fisher Axiom Genome-Wide CHB Array with customized contents in 2011, contained ~650,000 markers on the GRCh37 coordinates, providing comprehensive coverage of common genetic variation for GWASs. We used QCed and imputed genotype data in the mQTL analysis. First, we performed pre-imputation QC, removing variants and samples with a high missing rate. Principal component analysis (PCA) was conducted using the 1000 Genomes Project phase 3 (1KGP3) data as a reference panel to infer genetic ancestry and identify population outliers. Subsequently, we excluded samples exhibiting a heterozygosity rate beyond 6 standard deviations from the sample mean, as well as samples showing inconsistencies between genetic sex and self-reported sex. Within the EAS population, variants with low call rates or failing the Hardy–Weinberg equilibrium (HWE) test were discarded. After that, TWB genotypes were phased using Eagle v2.4 and imputed using Minimac4 with the 1KGP3 EAS data as the reference panel. In post-imputation QC, we removed variants with poor imputation quality (INFO score < 0.6) or low MAF (MAF < 0.005), yielding approximately 8 million genetic variants in the mQTL analysis.

**DNAm.** Following a minimum fasting period of 6 h, each participant enrolled in the TWB underwent physical examinations and provided blood and urine samples. Between 2016 and 2021, a total of 2474 TWB participants were randomly selected for DNAm measurement.

The Illumina Infinium MethylationEPIC BeadChip (Illumina, Inc., San Diego, CA) was employed to quantify DNAm values in the participants' blood samples. This high-throughput array covers approximately 860,000 CpG sites, allowing for a comprehensive assessment of DNAm patterns. The utilization of this platform enabled the acquisition of detailed DNAm data, facilitating the investigation of mQTLs and their potential associations with various phenotypic traits and diseases.

We performed QC of methylation in R using the chAMP package, which is designed to analyze Illumina Methylation beadarray data[27]. The raw data was loaded using the champ.load function, followed by automatically filtering out SNPs that might influence the analysis results. Then using the champ.filter function, we filtered: (1) probes with detection *P*-value > 0.01; (2) probes with less than 3 beads in at least 5% of the samples per probe; (3) all non-CpG probes; and (4) all multi-hit probes. 772,660 CpG set probes remained. We then used dasen[28] to normalize the type II bias. Batch and position effects were corrected by the champ.runCombat function. Cell-type proportions were estimated by the champ.refbase function and then corrected for in DNAm values to reduce confounding by cell-type heterogeneity.

To account for the remaining confounding effects that may arise from both known and unknown sources, we first used the PEER method to estimate latent factors influencing methylation data[29]. Additionally, we performed PCA on genotype data to capture the structure of genetic ancestry. Subsequently, we incorporated these factors (one PEER factor, explaining > 99% of variance, and 20 genotypic PCs) along with the known covariates (sex, age, and smoking status) into a linear regression model to adjust for their influence on methylation levels. Finally, the methylation data residuals from the linear regression model underwent rank-based inverse normal transformation (RINT) to achieve normality prior to mQTL analysis. We included 755,476 autosomal CpG probes in the mQTL analysis.

### mQTL mapping

We performed mQTL mapping using QTLtools (v1.2)[13] in 1,997 individuals with genotype and DNAm data. We tested the association between the CpG sites and genetic variants (with MAF > 0.5%) located within ±1 Mp of the CpG site in the study population (TWB).

## CpG heritability

We evaluated CpG heritability by extracting CpG site heritability estimates from a previously published twin study conducted in the NTR biobank[14,15]. We used heritability estimates from 364,047 CpG sites to compare the heritability of mCpGs with non-mCpGs.

## Enrichment analysis of mCpG sites in CpG categories and chromatin states

To investigate the enrichment of mCpG sites within specific CpG categories (such as CpG island, N_shore, N_shelf, S_shore, and S_shelf), we assessed the overlap between the mCpGs and CpG categories. The CpG categories were from the EPIC array annotation (see "Data availability" section). To assess the enrichment of mCpGs in each CpG category relative to non-methylated CpG sites, we performed a two-sided Fisher's exact test for each CpG category with Opea_sea as background. For each CpG category, OR was calculated as (number of mCpGs in the CpG category/number of non-CpGs in the CpG category)/(number of mCpGs in Open_sea/number of non-mCpGs in Open_sea).

To investigate the enrichment of mCpG sites within specific chromatin states, we assessed the overlap between the mCpGs and the Roadmap Chromatin state annotations[30]. The annotations were from the Roadmap Epigenomics Mapping Consortium, including 15 chromHMM states across 127 cell types. To assess the enrichment of mCpGs within each chromatin state relative to non-methylated CpG sites, (1) we calculated the OR = (number of mCpGs in the chromatin state/number of non-mCpGs in the chromatin state)/(number of mCpGs outside the chromatin state/number of non-mCpGs outside the chromatin state) for each cell type; (2) we computed the average log-transformed OR (logOR) and its standard deviation across all cell types; (3) statistical significance was evaluated using a two-sided $Z$-test, with a Bonferroni-corrected $P$-value threshold of $P < 0.003$ (0.05/15).

## mQTL effect size comparing

To evaluate the consistency of mQTL effect sizes across EAS studies and between the EAS and EUR populations, we used mQTLs identified in an independent South Asian (SA) mQTL study[10] as a benchmark, mitigating potential winner's curse effects. Specifically, we selected the top significant mQTL for each CpG from the SA study. We then retained only those mQTLs with a MAF > 0.05 across the EAS, SAS, and EUR populations (1KGP3) that were also present in the EAS_TWB, EAS_Hatton, EAS_Peng, and EUR_Min summary statistics. This process resulted in a final benchmark set of 20,926 mQTLs for the subsequent effect size comparison. For each comparison group, we computed the pairwise Pearson correlation coefficient and fitted a linear regression model. We standardized the effect size and standard error of the mQTL for EAS_Peng using the method described in ref. 31. The standardized EAS_Peng summary statistics were used for mQTL effect size comparing analysis and meta-analysis.

## Meta-analysis

For identifying mQTLs in EAS, we performed inverse variance fixed-effect meta-analysis to combine mQTL results from EAS_TWB, EAS_Hatton, and EAS_Peng using metal[32]. For identifying mQTLs across EAS and EUR, we performed inverse variance fixed-effect meta-analysis to combine mQTL results from EAS_TWB, EAS_Hatton, EAS_Peng, and EUR_Min using metal[32].

## Partitioned heritability analysis with LDSC

We performed LDSC to estimate the partitioned heritability of complex traits and diseases explained by mQTLs[33]. We obtained summary statistics for SCZ and IBD in the EAS population from the Psychiatric Genomics Consortium (PGC) and the International Inflammatory Bowel Disease Genetics Consortium (IIBDGC)[16,17], respectively. From the BBJ project, we obtained summary statistics for 220 traits and diseases[34]. We estimated their SNP-based heritability using LDSC. A total of 48 traits and diseases from BBJ with a SNP-based heritability greater than 0.03 were used for further heritability enrichment analysis.

We constructed mQTL annotations using the MaxCPP method[35]. Briefly, we performed fine-mapping for each mCpG site using the approximate Bayes factors fine-mapping method (coloc R package), assuming a single causal variant per mCpG. For each mCpG, we computed the causal posterior probability (CPP), also known as posterior inclusion probability (PIP), for each variant associated with the mCpG. For each variant, we then selected the maximum of the CPP (MaxCPP) across its associated CpG sites where this variant is within the 95% credible set. This MaxCPP value served as the mQTL annotation value in the subsequent LDSC analysis. Linkage disequilibrium (LD) scores were calculated using these MaxCPP annotations and genotype data from the EAS individuals in the 1KGP3. Both the mQTL annotations and all annotations from the baseline model were incorporated into the LDSC analysis.

To account for differences in array types (EPIC vs 450K) when comparing mQTL studies from EAS and EUR, we restricted the EAS mQTLs to include only CpG sites present on the 450K array. These filtered mQTLs, along with all annotations from the baseline model, were then incorporated into the LDSC analysis.

To assess how MAF differences may affect mQTL contribution to disease heritability across ancestries, we measured the conditional standardized effect size ($\tau_*$) of mQTLs using LDSC[35,36]. We (1) calculated the difference in MAF ($\Delta$MAF) as $\Delta$MAF = MAF_EAS - MAF_EUR for each variant, using MAF from 1KGP3 for EAS (MAF_EAS) and EUR (MAF_EUR); (2) incorporated $\Delta$MAF as an annotation in LDSC analysis, along with annotations from the baseline model, EAS mQTLs, and EUR mQTLs to estimate the conditional standardized effect size of EAS mQTLs ($\tau_*$_EAS) and EUR mQTLs ($\tau_*$_EUR); (3) calculated the difference in conditional standardized effect size of mQTLs between EAS and EUR ($\Delta\tau_*$) as $\Delta\tau_* = \tau_*$_EAS − $\tau_*$_EUR; (4) repeated (2) and (3) excluding $\Delta$MAF as an annotation in the LDSC analysis to assess the conditional standardized effect size of EAS mQTLs and EUR mQTLs without adjusting for $\Delta$MAF.

## Colocalization analysis

We performed multiple traits colocalization analysis using HyPrColoc[37] to assess colocalization between mCpGs (identified from EAS_meta) and 224 complex traits or diseases from BBJ and EAS GWAS (SCZ and IBD). In the HyPrColoc analysis, we used summary statistics from EAS_meta and traits from BBJ and EAS GWAS filtered at a $P$-value threshold of 0.01. A posterior probability of colocalization greater than 0.7 was considered statistically significant.

## SMR analysis

We used an SMR approach to investigate whether CpG methylation levels causally influence various traits[31]. Briefly, summary statistics for 224 traits were obtained from BBJ and EAS GWAS (SCZ and IBD). LD information was estimated using genotype data from the TWB. In the SMR analysis, we selected mQTL variants identified in the EAS_meta as instrumental variables (IVs) by the default threshold of $P = 5e$-8. Furthermore, we performed a heterogeneity test using the HEIDI test. Only CpG-trait associations with both a significant SMR $P$-value < 5e-10 (0.05/70,716,988, Bonferroni-corrected) and HEIDI $P$-value > 0.05 were considered evidence for a causal effect.

## Gene set enrichment analysis

Gene set enrichment analysis was performed using MAGMA[22]. Briefly, genetic variants were annotated to genes using RefSeq NCBI37.3, with a 50 kb window around each gene. Summary statistics for the traits and diseases were obtained from BBJ and EAS GWAS (SCZ and IBD). LD estimates were derived from the 1KGP3 EAS population. For gene-level

analysis, we calculated gene association *P*-values for traits that exhibited significant pleiotropic effects mediated by mCpGs (identified through SMR and colocalization analyses) using the SNPwise-mean model in MAGMA. For gene-set analysis, gene ontology (GO) biological process gene sets were obtained from the Molecular Signatures Database (MSigDB). For each CpG-trait cluster identified by HyPrColoc, we performed gene set enrichment analysis for each trait within that cluster using MAGMA with default settings. This analysis specifically focused on gene sets that included the gene potentially regulated by the mCpG (the nearest gene to the mCpG). Multiple testing correction was applied using the Bonferroni method, with significance defined as $P < 0.05/$(number of gene sets × number of traits).

## Reporting summary

Further information on research design is available in the Nature Portfolio Reporting Summary linked to this article.

## Data availability

Individual-level genotype data and methylation data from the Taiwan biobank are available at https://www.biobank.org.tw/english.php; mQTL Summary statistics for Taiwan biobank (EAS_TWB), EAS_meta and EAS + EUR_meta are available at https://lin-lab.site/data/; mQTL summary statistics from EAS_Hatton are available at https://yanglab.westlake.edu.cn/software/smr/#mQTLsummarydata; mQTL summary statistics from EAS_Peng are available at https://www.biosino.org/node/project/detail/OEP002902; mQTL summary statistics from EUR_Min are available at http://mqtldb.godmc.org.uk/downloads; mQTL summary statistics from South Asian mQTL study are available at https://zenodo.org/record/5196216#.YRZ3TfJxeUk; GWAS summary statistics for 220 traits from BBJ are available at https://pheweb.jp/downloads; GWAS summary statistics for IBD in EAS are available at https://www.ibdgenetics.org; GWAS summary statistics for SCZ in EAS are available at https://pgc.unc.edu/for-researchers/download-results/; 1000 Genomes Project Phase 3 is available from https://www.internationalgenome.org/category/phase-3/; CpG annotation is available from https://zhouserver.research.chop.edu/InfiniumAnnotation/20180909/EPIC/EPIC.hg19.manifest.tsv.gz; CpG categories annotation is available from https://support.illumina.com/downloads/infinium-methylationepic-v1-0-product-files.html; Chromatin, Transcription factor binding sites, Imprinting control regions, and CTCF binding sites annotation data for CpG is available from http://zwdzwd.github.io/InfiniumAnnotation#current; GO gene sets from MSigDB v2025.1 are available at https://www.gsea-msigdb.org/gsea/msigdb; Source data are provided with this paper.

## Code availability

plink 1.9beta: https://www.cog-genomics.org/plink/. Eagle v2.4: https://alkesgroup.broadinstitute.org/Eagle/. Minimac4: https://genome.sph.umich.edu/wiki/Minimac4. QTLtools 1.3.1: https://qtltools.github.io/qtltools/. chAMP: https://bioconductor.org/packages/release/bioc/html/ChAMP.html. PEER: https://www.sanger.ac.uk/tool/peer/. LDSC v1.0.1: https://github.com/bulik/ldsc. METAL v2011-03-25: https://genome.sph.umich.edu/wiki/METAL. Coloc v5: https://chr1swallace.github.io/coloc/. HyPrColoc 0.0.2: https://github.com/jrs95/hyprcoloc. SMR: https://yanglab.westlake.edu.cn/software/smr/. MAGMA v1.10: https://cncr.nl/research/magma/

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

## Acknowledgements

We thank the participants of Taiwan Biobank and the staff, management team, and leadership of Taiwan Biobank. This research has been conducted using the Taiwan Biobank Resource under application number TWBR11111-10. We would like to thank National Core Facility for Biopharmaceuticals (NCFB, 113-2740-B-492-001) and National Center for High-performance Computing (NCHC) of National Applied Research Laboratories (NARLabs) of Taiwan for providing computational resources and storage resources. This study was supported by the National Health Research Institutes (NP-109, 110, 111, 112, 113, 114-PP-09 to Y.-F.L.), the Ministry of Science and Technology (MOST 109-2314-B-400-017, 110-2314-B-400-028-MY3 to Y.-F.L.), and the National Science and Technology Council (NSTC 113-2628-B-400-002 to Y.-F.L.), Taiwan. H.H. acknowledges support from the Stanley Center for Psychiatric Research, Merkin Institute Fellowship, and the Zhengxu and Ying He Foundation. Y.-C.A.F. acknowledges support from the NSTC (112-2314-B-002-200-MY3) and the Ministry of Education (the Yushan Young Fellow Program, MOE-111-YSFAG-0003-001-P1).

## Author contributions

C.-Y.C., Y.-C.A.F., H.H., and Y.-F.L. designed the study and supervised the work. R.L., T.-T.C., Y.X., S.L., and T.G. analyzed the data and helped in study management. R.L., T.-T.C., Y.X., C.-Y.C., Y.-C.A.F., H.H., and Y.-F.L. wrote the manuscript.

## Competing interests

Chia-Yen Chen is an employee of Merck & Co., Inc. Other authors declare no competing interests.
