## [Transparent Peer Review file · Nature Communications]

Genetic regulation of methylation across East Asian and European populations

Corresponding Author: Professor Hailiang Huang

Version 0:

Reviewer comments:

Reviewer #1

(Remarks to the Author)

Manuscript by Liu et al reports the largest to date methylation quantitative trait loci (mQTL) study in Han Chinese. They used three cohorts, totaling 7,620 individuals, to identify and replicate/meta-analyze mQTLs (they refer to CpGs that are genetically determined as mCpGs). Newly collected DNA methylation data in the Taiwan biobank cohort (N=1,998) are used for discovery and two published mQTL studies in Han Chinese (N=2,099 for Hatton et al and 3,523 for Peng et al) are used for replication. They then compared the mQTLs with those in Europeans and performed analyses to demonstrate the utility of mQTLs in interpreting GWAS results and investigated CpG-trait associations both at individual mCpG-trait pairs (SMR) and clusters of traits influenced by shared causal variants at a given CpG (hyPrColoc). They highlight a couple of examples: cg10771262, located in the promoter region of TCF21, exhibiting pleiotropic effects on CAD, and three CpG sites in the promoter of CAMK1D (cg10704395, cg03575602, and cg14427642) and a group of T2D-related traits.

This is an important study and resource for genetic studies in East Asian populations, therefore, of interest to the broad readership of Nature Communications. Key strengths of the manuscript are rigorous analyses, well-presented and interpreted results, and appropriate discussion of the contribution this study makes to the field.

I only have one suggestion for improvement. I think it is important to be clearer about replication results. When it is stated that “95% of our identified mCpGs were also reported as mCpGs in at least one of the previous studies”, this high number must be driven by the Peng study that has 850k data as opposed to the Hutton study that is on the 450k platform and is therefore missing a number of CpGs. Breaking down these numbers would be helpful. Otherwise, this manuscript represents an excellent study and reporting of the results.

Reviewer #2

(Remarks to the Author)

This manuscript presents a large-scale, multi-ancestry mQTL meta-analysis integrating East Asian and European populations. The authors expand the East Asian mQTL resource (N = 7,620) and perform a cross-population meta-analysis with European data (N = 27,750), further integrating GWAS summary statistics for colocalization analysis. The overall data quality is high, and the analytical pipeline is rigorous. This work provides a valuable resource for future methylation QTL studies, especially in East Asian populations. The authors adopted a standardized analysis framework, utilizing established tools such as QTLtools, SMR, HyPrColoc, and LDSC, with appropriate quality control measures. The manuscript is clearly written, and the results are of practical interest to researchers focusing on GWAS fine mapping and population-specific regulatory analysis. But some concerns need to be addressed:

1. The authors report the identification of ~30,000 novel mCpG sites. However, the functional relevance of these novel sites is unclear. Additional analyses to characterize their regulatory potential or biological significance would strengthen the impact of the findings.
2. The study observes significant heritability enrichment of EAS mQTLs for schizophrenia and inflammatory bowel disease, but not for EUR mQTLs. This population-specific enrichment is potentially important, but the underlying explanation is lacking and requires further elaboration.

3. The colocalization analysis with BBJ phenotypes identifies pleiotropic signals between mCpGs and traits such as coronary artery disease (e.g. angina, myocardial infarction), related medications (e.g. vasodilators, aspirin), and leukocyte counts. However, the biological interpretation of these results remains superficial. The authors are encouraged to incorporate functional validation, pathway enrichment analysis, or discussions of phenotypic antagonism to better contextualize these associations.

4. The role of minor allele frequency (MAF) differences in shaping population-specific mQTLs has already been well documented. This manuscript reiterates that point but does not offer novel mechanistic insights into how these differences impact regulatory architecture.

5. On page 20, lines 383–384, the authors state: “Our analysis focused on cis-mQTLs due to the unavailability of trans-mQTL summary statistics for the EAS_Hatton and EAS_Peng datasets.” This statement appears to be inaccurate. At least the EAS_Peng dataset includes trans-mQTL data, as indicated by the data availability link provided (<https://www.biosino.org/node/project/detail/OEP002902>). This should be corrected.

6. Section 6, titled “Cross-population mQTL meta-analysis,” is underdeveloped. It only presents the meta-analysis results without any interpretation or integration with prior analyses. For clarity and narrative coherence, this section could be merged with the earlier EAS meta-analysis section.

7. The authors are strongly encouraged to publicly release the summary statistics from both the three-EAS-population meta-analysis and the combined EAS+EUR meta-analysis. This would greatly increase the utility of the dataset for researchers conducting GWAS fine-mapping and investigating ancestry-specific regulatory mechanisms.

Version 1:

Reviewer comments:

Reviewer #1

(Remarks to the Author)

The authors did an excellent job addressing all reviewer comments.

Reviewer #2

(Remarks to the Author)

Most of my previous concerns have been well addressed. I have only one suggestion for the authors' consideration.

While I understand that one of the datasets (N = 2,099) did not include trans-mQTLs, the other two datasets (N = 1,998 and N = 3,523) did provide trans-mQTL information. Given that trans-mQTLs are also valuable resources for understanding distal regulatory effects, I encourage the authors to consider making the Taiwan Biobank and meta-level trans-mQTL datasets publicly available for future research.

REVIEWER COMMENTS

Reviewer #1 (Remarks to the Author):

Manuscript by Liu et al reports the largest to date methylation quantitative trait loci (mQTL) study in Han Chinese. They used three cohorts, totaling 7,620 individuals, to identify and replicate/meta-analyze mQTLs (they refer to CpGs that are genetically determined as mCpGs). Newly collected DNA methylation data in the Taiwan biobank cohort (N=1,998) are used for discovery and two published mQTL studies in Han Chinese (N=2,099 for Hatton et al and 3,523 for Peng et al) are used for replication. They then compared the mQTLs with those in Europeans and performed analyses to demonstrate the utility of mQTLs in interpreting GWAS results and investigated CpG-trait associations both at individual mCpG-trait pairs (SMR) and clusters of traits influenced by shared causal variants at a given CpG (hyPrColoc). They highlight a couple of examples: cg10771262, located in the promoter region of TCF21, exhibiting pleiotropic effects on CAD, and three CpG sites in the promoter of CAMK1D (cg10704395, cg03575602, and cg14427642) and a group of T2D-related traits.

This is an important study and resource for genetic studies in East Asian populations, therefore, of interest to the broad readership of Nature Communications. Key strengths of the manuscript are rigorous analyses, well-presented and interpreted results, and appropriate discussion of the contribution this study makes to the field.

We are glad that the reviewer enjoyed this study and found it important and clearly and appropriately presented!

I only have one suggestion for improvement. I think it is important to be clearer about replication results. When it is stated that “95% of our identified mCpGs were also reported as mCpGs in at least one of the previous studies”, this high number must be driven by the Peng study that has 850k data as opposed to the Hutton study that is on the 450k platform and is therefore missing a number of CpGs. Breaking down these numbers would be helpful. Otherwise, this manuscript represents an excellent study and reporting of the results.

We agree with the reviewer and have done the corresponding analysis. As expected, 93.89% of EAS_TWB mCpG were reported in EAS_Peng. Among them, 53.18% were exclusively detected by the 850k EPIC array. In contrast, 37.18% of EAS_TWB were reported in EAS_Hatton, and all of these are present on both EPIC and 450K arrays. A detailed comparison of mCpGs across all EAS studies is presented in "Figure S1. Comparison of mCpG across EAS studies."

We have added these new results to the manuscript (Page 7).

Reviewer #2 (Remarks to the Author):

This manuscript presents a large-scale, multi-ancestry mQTL meta-analysis integrating East Asian and European populations. The authors expand the East Asian mQTL resource (N = 7,620) and perform a cross-population meta-analysis with European data (N = 27,750), further integrating GWAS summary statistics for colocalization analysis. The overall data quality is high, and the analytical pipeline is rigorous. This work provides a valuable resource for future methylation QTL studies, especially in East Asian populations. The authors adopted a standardized analysis framework, utilizing established tools such as QTLtools, SMR, HyPrColoc, and LDSC, with appropriate quality control measures. The manuscript is clearly written, and the results are of practical interest to researchers focusing on GWAS fine mapping and population-specific regulatory analysis. But some concerns need to be addressed:

Thank you for the positive feedback and the very helpful suggestions below!

1. The authors report the identification of ~30,000 novel mCpG sites. However, the functional relevance of these novel sites is unclear. Additional analyses to characterize their regulatory potential or biological significance would strengthen the impact of the findings.

We thank the reviewer for this great question. In response, we performed additional analyses to characterize the functional impact of the novel mCpG sites.

Specifically, we investigated the potential function of these novel mCpG sites within key regulatory elements, including CTCF binding sites, transcription factor binding sites (TFBS), various chromatin states, imprinted regions, and DNase Hypersensitivity Sites (DHSs). We reported two key observations: 1) the novel mCpGs exhibit regulatory patterns highly consistent with those of previously reported mCpGs; 2) a subset of these novel mCpGs locates within regulatory regions that were not previously reported to be influenced by mQTLs. These additional analyses showed the potential functional impact of the newly identified mCpGs, which may enhance our understanding of gene regulation mechanisms.

We have presented the details on Pages 10-11 and in Figure S8.

2. The study observes significant heritability enrichment of EAS mQTLs for schizophrenia and inflammatory bowel disease, but not for EUR mQTLs. This population-specific enrichment is potentially important, but the underlying explanation is lacking and requires further elaboration.

We apologize for the lack of clarity in our earlier manuscript. We actually performed the heritability enrichment analyses for traits in BBJ, schizophrenia, and inflammatory bowel diseases, and reported the findings collectively. We have now clarified these results by reporting them individually.

Our approach leveraged the conditional LDSC framework to assess the impact of MAF differences between EAS and EUR populations on the per-SNP heritability explained by mQTLs. We found that the differences in heritability enrichment of mQTLs between EAS and EUR are largely driven by the differences in MAF.

Specifically, when conditioning on ΔMAF (the difference in MAF between EAS and EUR) in our LDSC model, we observed a reduction in $\Delta\tau^*$ (the difference in per-SNP heritability explained by EAS mQTLs versus EUR mQTLs) for these traits/diseases, demonstrating that MAF differences significantly contribute to the population-specific SNP heritability enrichment for mQTLs. For instance, $\Delta\tau^*$ decreased from 3.9 to 3.6 for CD, 1.19 to 0.4 for UC, 3.4 to 3.2 for IBD, and 0.97 to 0.2 for SCZ after accounting for the impact of MAF differences (p-value = 0.03, paired t-test).

This suggests that genetic variants with highly differentiated MAFs between EAS and EUR underlie differences in mQTL detection, thereby explaining the observed difference in heritability explained by mQTLs.

Our revised results and further discussion on this analysis can be found on Pages 12-13.

3. The colocalization analysis with BBJ phenotypes identifies pleiotropic signals between mCpGs and traits such as coronary artery disease (e.g. angina, myocardial infarction), related medications (e.g. vasodilators, aspirin), and leukocyte counts. However, the biological interpretation of these results remains superficial. The authors are encouraged to incorporate functional validation, pathway enrichment analysis, or discussions of phenotypic antagonism to better contextualize these associations.

We appreciate this suggestion. To further explore the biological mechanisms of pleiotropic signals, we attempted to identify potentially overlapping biological pathways across traits within the CpG-trait clusters identified through colocalization. We performed gene set enrichment analysis using MAGMA for each trait within the CpG-trait clusters. The gene set enrichment analysis leveraged the GO biological process (BP) gene set definitions from MSigDB (v2025.1), and we only included gene sets relevant to the target CpG-trait cluster, defined as those containing the gene nearest to the mCpG, under the assumption that mCpG often regulates the nearest gene. We found that 17.5% of the CpG-traits pleiotropic clusters had at least one GO BP gene set significantly enriched in more than one trait. For example, for the pleiotropic effects of cg10771262 (nearest gene *TCF21*) on CAD, we found the vasculature development GO BP gene set (Renal system vasculature development, Bonferroni-corrected P-value < 0.05) significantly enriched for four of six traits in the pleiotropic group: Angina, MI, SAP, and ATC_C01D (Figure S11). This suggests that cg10771262, through regulating *TCF21*, may mediate its pleiotropic effects on these CAD-related traits through its role in vascular development.

The revised methods, results and discussion for this analysis can be found on Pages 31, 20-21 and 22-23, Figures S11 and S12, and Table S7.

4. The role of minor allele frequency (MAF) differences in shaping population-specific mQTLs has already been well documented. This manuscript reiterates that point but does not offer novel mechanistic insights into how these differences impact regulatory architecture.

We agree with the reviewer that the role of MAF differences in shaping population-specific mQTLs has already been well documented. Respectfully, we did not relate this to mechanistic insights because we believe this role largely reflects the statistical power rather than biology.

Specifically, our analysis demonstrates that the power to identify population-specific mQTLs is substantially influenced by MAF differences across ancestries (Figure S5). A genetic variant that is common (e.g., $MAF > 5\%$) in one population but absent ($MAF = 0\%$) in another will lead to a population-specific mQTL discovery, as it can only be statistically tested for its regulation of methylation in the former group. Achieving sufficient statistical power for mQTL detection requires considerably larger sample sizes in populations where the variant's MAF is lower. For instance, a variant with an effect size of 1 standard deviation unit on methylation, with an MAF of 0.05 in EAS and 0.001 in EUR, would require only 690 samples in EAS for 95% detection probability at a P-value threshold of $1e-11$. In contrast, achieving the same power in EUR would require 35,680 samples, 52.3 times more than in EAS (Figure S13). Our empirical observations align with this expectation, showing that EAS-specific mQTLs are frequently linked to variants with significantly lower MAFs in EUR, making their detection challenging even in larger EUR cohorts (Figure S5).

While biological mechanisms are shared across populations (Figure S6), population-specific mQTLs are essential for accurately interpreting GWAS findings from diverse ancestries. We found that MAF differences across populations explain why the contribution of mQTLs to disease heritability does not transfer well across populations (see our response to comment #2). This underscores the critical importance of conducting adequately powered mQTL studies in diverse ancestral groups to fully characterize the genetic regulation of DNA methylation and its implications for complex traits across populations.

The revised discussion can be found on Page 22 and Figure S13.

5. On page 20, lines 383–384, the authors state: “Our analysis focused on cis-mQTLs due to the unavailability of trans-mQTL summary statistics for the EAS_Hatton and EAS_Peng datasets.” This statement appears to be inaccurate. At least the EAS_Peng dataset includes trans-mQTL data, as indicated by the data availability link provided (<https://www.biosino.org/node/project/detail/OEP002902>). This should be corrected.

We apologize for this inaccuracy. We have revised this sentence:

“Our analysis focused on cis-mQTLs because trans-mQTL was not available in one of the constituent studies (EAS_Hatton).”

6. Section 6, titled “Cross-population mQTL meta-analysis,” is underdeveloped. It only presents the meta-analysis results without any interpretation or integration with prior analyses. For clarity and narrative coherence, this section could be merged with the earlier EAS meta-analysis section.

We agree with the reviewer and have merged this section with the EAS meta-analysis section. Furthermore, we have included analyses characterizing the biological function of the novel mCpGs within this revised section.

The revised results can be found on Pages 10-11.

7. The authors are strongly encouraged to publicly release the summary statistics from both the three-EAS-population meta-analysis and the combined EAS+EUR meta-analysis. This would greatly increase the utility of the dataset for researchers conducting GWAS fine-mapping and investigating ancestry-specific regulatory mechanisms.

We agree with the reviewer and will release the summary statistics of EAS_meta and EAS+EUR_meta along with EUR_TWB at <https://lin-lab.site/data/>.

Reviewer #1 (Remarks to the Author):

The authors did an excellent job addressing all reviewer comments.

We are delighted to hear this. Thank you so much again for your very helpful comments.

Reviewer #2 (Remarks to the Author):

Most of my previous concerns have been well addressed. I have only one suggestion for the authors' consideration.

While I understand that one of the datasets (N = 2,099) did not include trans-mQTLs, the other two datasets (N = 1,998 and N = 3,523) did provide trans-mQTL information. Given that trans-mQTLs are also valuable resources for understanding distal regulatory effects, I encourage the authors to consider making the Taiwan Biobank and meta-level trans-mQTL datasets publicly available for future research.

We completely agree. We have begun working on the trans-mQTL analysis and will publicly release the trans-mQTL summary statistics in the same location as the cis-mQTL data. A README file describing the methods used in the trans analysis will be included. Given the computational scale, we anticipate that this will take a few months to complete. Our goal is to make this a community resource, before or even without a separate publication, similar to the Neale Lab UK Biobank GWAS resource (<https://www.nealelab.is/uk-biobank>).